# Milk Authentication: Stable Isotope Composition of Hydrogen and Oxygen in Milks and Their Constituents

**DOI:** 10.3390/molecules25174000

**Published:** 2020-09-02

**Authors:** Staša Hamzić Gregorčič, Doris Potočnik, Federica Camin, Nives Ogrinc

**Affiliations:** 1Department of Environmental Sciences, Jožef Stefan Institute, Jamova 39, 1000 Ljubljana, Slovenia; stasa.gregorcic@ijs.si (S.H.G.); doris.potocnik@ijs.si (D.P.); 2Jožef Stefan International Postgraduate School, Jamova 39, 1000 Ljubljana, Slovenia; 3Department of Food Quality and Nutrition, Research and Innovation Centre, Fondazione Edmund Mach, via Mach 1, 38010 San Michele all’Adige, Italy; federica.camin@fmach.it; 4Center Agriculture Food Environment (C3A), University of Trento, via Mach 1, 38010 San Michele all’Adige (TN), Italy

**Keywords:** milk, adulteration, water addition, oxygen stable isotopes, lactose

## Abstract

This paper summarises the isotopic characteristics, i.e., oxygen and hydrogen isotopes, of Slovenian milk and its major constituents: water, casein, and lactose. In parallel, the stable oxygen isotope ratios of cow, sheep, and goat’s milk were compared. Oxygen stable isotope ratios in milk water show seasonal variability and are also ^18^O enriched in relation to animal drinking water. The *δ*^18^O_water_ values were higher in sheep and goat’s milk when compared to cow milk, reflecting the isotopic composition of drinking water source and the effect of differences in the animal’s thermoregulatory physiologies. The relationship between *δ*^18^O_milk_ and *δ*^18^O_lactose_ is an indication that even at lower amounts (>7%) of added water to milk can be determined. This procedure once validated on an international scale could become a reference method for the determination of milk adulteration with water.

## 1. Introduction

After the melamine milk powder incident in China in 2008 the adulteration of milk and dairy products highlighted the need for greater transparency in the food chain, guarantees surrounding food quality and safety and the development of methods for determining the authenticity of dairy products [1]. Although milk is a frequent target for fraud [2], available knowledge and data about methods for the prevention or mitigation of the fraud issue is still limited. In order to assure the authenticity of milk, one requires a deep understanding of the characteristics of authentic milk. In response, scientists have developed new analytical techniques and strategies [3,4], which will assist milk producers and suppliers in the detection and prevention of milk fraud.

When milk is diluted with water, its nutritional value decreases and in addition chemicals are added to compensate the density and the colour after dilution, thus posing a potential risk to human health [2]. Further, there are very strong economic arguments of minimizing the allowed amount of added water to milk, since the price of milk is based on milk solids contents. The processing of milk also provides an opportunity for producers to add water beyond the acceptable limits in preserved milk, which is illegal. Since it is not compulsory to state the amount of added water on the label, some companies take advantage of this legal loophole.

Several methods to detect adulterants in milk exists including measurement of freezing point depression, electrical admittance spectroscopy, single-frequency conductance measurements, digital image chromatography, ultraviolet (UV) visible light spectroscopy, and enzyme linked immunosorbent assay [5,6,7]. Determining the milk water content is typically performed using traditional methods such as by measuring changes in freezing point of the milk or changes in the refraction of light through the whey component of milk after precipitation and removal of the casein and fat components using either acetic acid or copper sulfate. Current methods can be classed as direct contact methods, which are not reliable for making continuous measurements. Other methods involve separating the water from milk solids and then quantifying the amount of water by weight or volume—these techniques, although accurate, are time-consuming and expensive. Among modern techniques near infrared (NIR) spectroscopy has proved to be a fast non-destructive method for food safety evaluation and control [8,9,10,11], and can be also used to detect water and its content in milk [10]. The main drawback is that the milk has a near-infrared absorption spectrum similar to that of the water [9]. Time-domain nuclear magnetic resonance (TD-NMR) method has been used for quantification of fat and water content in cheese [12] and to identify several adulterants in milk such as water, whey, synthetic milk, synthetic urine and hydrogen peroxide [13]. Although the method is widely used in dairy studies, it has some restrictions, especially in samples with either low water or low-fat concentration (<5% *v*/*v*).

The use of stable isotopes of light elements is an approach of a grown interest in terms to discover possible commercial fraud [14]. Several studies have demonstrated that stable oxygen isotope ratios (*δ*^18^O values) has been successfully applied to detect illegal watering of different types of food matrices such as wine [15], fruit juices [16], and concentrated spirits [17]. *δ*^2^H and *δ*^18^O values in water can provide key information on water origins (e.g., local precipitation, groundwater), climate (ambient temperatures during condensation and precipitation) and the degree of evapotranspiration [18,19,20]. The relationship between *δ*^2^H and *δ*^18^O values in the hydrosphere throughout the continents known as the meteoric water line (MWL; *δ*^2^H = 8 *δ*^18^O + 10) was first defined by Craig [21]. Besides the ‘latitude’ effect, there is a ‘continental’ effect due to the distance from the sea, related to the vapour masses moving over continents leading to the lower *δ*^2^H and *δ*^18^O values in precipitation (mean decrease of −2.8‰/1000 km from the coast). Moreover, different altitudes inland also lead to decrease in *δ*^2^H and *δ*^18^O values in precipitation since at higher altitude there is isotopically lighter vapour. Finally, another variation in *δ*^2^H and *δ*^18^O values can occur due to seasonal trends; during summer the enrichment in ^2^H and ^18^O in precipitation, especially inland, occurs.

The sources of H and O in animals are drinking water, food, food water and in case of O also molecular O_2_ [22]. Groundwater the main source of animal drinking water has an isotopic composition depending on geographical factors such as altitude, latitude and distance from the sea, but not on the season. In plants, the main components of feed, the isotopic composition of water are positive relative to those of the corresponding soil water. Furthermore, the *δ*^18^O values in plants reflect evaporative enrichment transpiring leaves and isotopic exchange between plant water and organic molecules [23,24]. The average *δ*^18^O value of the body water of most domestic animals is about 3 ± 1‰ more positive than that of the drinking water [25]. Consequently, the enrichment in ^2^H and ^18^O was observed also in milk where the metabolism, and isotopic fractionation during milk synthesis cause additional isotopic fractionation. Overall, the isotopic composition of milk depends on species, drinking, and respiration rates [25], season, farm conditions, breed, and the physiological condition of the animal [26,27]. Dairy animal species with different thermoregulatory physiology should have different water isotope fractionation in body fluids, which is related to evaporation, as vapour is more depleted in heavy isotopes than other body fluids [28,29]. Further, goat milk has a higher proportion of calcium compared to cow milk, which is linked to the higher metabolic rate of the smaller animal [30]. Likewise, according to Bryant and Froelich [30] and Podlesak et al. [31], body surface area relative to body mass makes a mammal prone to water loss via evaporation. A relationship between *δ*^18^O in milk water and the season was reported by Kornexl et al. [32], due to seasonal changes in the *δ*^18^O of forage plants, as well as in the body of the animal, linked to evapotranspiration. *δ*^2^H and *δ*^18^O stable isotopes in milk were also used to detect its geographical origin, due to the relationship between the isotopic signature of milk and that of the drinking water of regions located at different latitudes and/or altitudes [33,34]. More recently, the exchange of H and O between organic molecules and animal’s body water due to metabolism and biosynthesis were studied. The results suggested that H isotopes carry a signature related to dietary habits of the animal, while O isotopic signature reflects more animal’s physiological water balance [35].

Our paper introduces the concept of using *δ*^2^H and *δ*^18^O measurements in the milk and its constituents as a natural isotopic toolbox to provide information about the sources of water in milk and to detect possible adulteration of milk with water. Thus, the main objectives of our study were to: (1) identify the differences in the *δ*^18^O value in milk according to the season, region and animal species; (2) identify the correlations between *δ*^18^O values in milk and drinking water; and (3) to test the use of *δ*^18^O values in lactose as an internal standard for the detection of water addition.

## 2. Results and Discussion

### 2.1. Stable Isotope Composition of Milk and Casein: Year, Season, Region and Species Variability

The *δ*^18^O_milk,_
*δ*^18^O_casein_ and *δ*^2^H_casein_ values of raw cow’s milk from farms in four geographical regions in Slovenia: Alpine, Dinaric, Pannonian, and the Mediterranean, broken down according to season and year of production are presented in Appendix A. The *δ*^18^O_milk_ values in collected cow milk samples (*n* = 319), produced between 2012 and 2015, ranged from −9.2‰ to −0.04‰ (Figure 1a–d). The *δ*^18^O_casein_ values ranged from 8.8‰ to 14.6‰, and the *δ*^2^H_casein_ values were from −150‰ to −100‰.

After applying an ANOVA test, significant differences (*p* < 0.05) in the *δ*^18^O_milk_ values according to region, year, and season were observed. The *δ*^18^O_milk_ values were higher in summer and in 2012 compared to 2013, 2014 and 2015. The results of the Tukey contrasts test (*p* < 0.05) indicate that the *δ*^18^O_milk_ values were the highest in the Mediterranean region, which out of the four regions has the mildest climate. Our findings are consistent with previous studies that show a seasonal variation in the *δ*^18^O values in milk water with higher ^18^O content in the summer milk [32,34,36,37]. This increase results from the high evapotranspiration rate in fresh plant feed and animals during the summer. The use of water isotopes as an indicator of the geographical origin of milk is, however, only useful if the type of feed is known (i.e., fresh grass vs silage) [35], which unfortunately was not the case in our study.

From Figure 2, it is evident that the casein was ^18^O-enriched by approximately 17‰ relative to the milk water, and both *δ*^2^H_casein_ and *δ*^18^O_casein_ values were consistent, although their values varied slightly from region to region in 2014 (Figure 2; Appendix A). No regional differences in *δ*^2^H_casein_ and *δ*^18^O_casein_ values were observed in 2013. Conversely in 2014, both the Mediterranean (*δ*^18^O_casein_ = 12.8 ± 1.3‰) and Pannonian (*δ*^18^O_casein_ = 12.2 ± 1.3‰) regions differ significantly from the Alpine (*δ*^18^O_casein_ = 11.4 ± 0.8‰) and Dinaric (*δ*^18^O_casein_ = 11.2 ± 0.7‰) ones (Figure 2). The highest *δ*^18^O_casein_ values were in milk produced at lower altitudes closer to the coast where the climate is dry and hot (Mediterranean region). Also, there were no significant regional differences in the average *δ*^2^H_casein_ values. It is interesting to note, that winter samples from the Dinaric region had a higher mean value *δ*^2^H_casein_ (−127‰) compared to the summer samples (−134‰).

No correlation was observed between *δ*^2^H_casein_ and *δ*^18^O_casein_ values (Figure 3), which supports the finding from previous studies that 30% of the H and 70% of the O in milk protein derives from the local water, with the remaining fraction originating from the diet. Also, it is necessary to consider possible sources of variation related to isotopic fractionation in the animal’s body water [29]. Similarly, the *δ*^2^H_casein_ values are influenced by the continuous exchange of ^2^H between the animal’s body water and drinking water in a specific location over time [30,31]. Thus, compared to the *δ*^18^O_milk_ values, *δ*^18^O_casein_ values provide a more consistent isotopic signature with which to determine the authenticity and origin of the milk.

Given the interest in detecting commercial fraud of milk and dairy products, we determined the *δ*^18^O_milk_ values in cow, sheep and goat milk collected from farms located in Mediterranean region (Brkini, Vipava), Dinaric (Karst) and Alpine region (Bovec) from May to June in 2012 and 2013. The *δ*^18^O_milk_ values in goat, sheep and cow ranged from −3.6 to 2.4‰, from −5.6 to 1.2‰ and from −6.6 to −2.6‰, respectively. 

It is evident from Figure 4 that *δ*^18^O_milk_ values in goat (average values: *δ*^18^O_milk_ = −0.9 ± 2.1‰ and *δ*^18^O_milk_ = −1.8 ± 1.0‰, in 2012 and 2013, respectively) and sheep milk (average values: *δ*^18^O_milk_ = −2.4 ± 1.6‰ in 2012 and *δ*^18^O_milk_ = −3.1 ± 1.6‰ in 2013) are higher than the values in cow milk (average values: *δ*^18^O_milk_ = −3.0 ± 0.5‰ in 2012; *δ*^18^O_milk_ = −5.0 ± 0.7‰ in 2013). First, such differences could be related to the source of drinking water. Comparing to cows that predominantly drink groundwater, the sources of drinking water for goats and sheep are also rainwater and grazing on fresh pasture herbage that is enriched in ^18^O. Another explanation for the isotopic difference is animal physiology and diet. Bryant and Froelich [30] proposed that herbivore oxygen isotope composition in water body depends principally on body size. Total water flux (amount of water into and out of animals each day) also scale with body size but can be also influenced by dietary inputs and environmental temperature. Larger animals might on average be less capable to conserving water compared to smaller animals, however difference in water conservation depends also on water consumption. For example, goat drink water every few days, while cows must drink water every day [29]. Thus, it is expected that goat with lower water turnover rate have higher *δ*^18^O_milk_ values. Finally, because sweat, urine, and fecal water have higher *δ*^18^O values than water vapor, animals that pant to lose heat (goat, sheep), have high urinary salt concentrations, and have low fecal water contents, should have a higher *δ*^18^O_milk_ values than animals that lose more of their water as liquid (cow) [27,29].

Further, *δ*^18^O_milk_ values in all three species are higher in 2012 comparing to 2013. One of the reasons could be unusual weather conditions in May and June in 2012 with extremely high temperatures (average: 14.1 and 20.6 °C, respectively) comparing to 2013 (average: 13.5 and 18.2 °C, respectively) that can influence the source of water as well as activity level and body temperature regulation [35]. Rapid metabolism and more intense respiration also likely cause evaporative ^18^O-enrichment in body water. Lower *δ*^18^O_milk_ values were also observed in the Alpine region connected to higher altitude, lower temperatures and higher amount of precipitation.

### 2.2. Stable Isotope Composition of Oxygen in Milk and Groundwater: Detection of Dilution with Water

Overall, *δ*^18^O_milk_ values in milk depend on the sources of drinking water, metabolism, and isotopic fractionation during milk synthesis. In most cases, drinking water is taken from local groundwater (GW) sources, which reflects the isotopic composition of the mean annual precipitation [18]. For example, Liu et al. [38] found that *δ*^2^H and *δ*^18^O values in goat milk water were identical to that in drinking water. The *δ*^2^H and *δ*^18^O values of the water in cow milk correlate with geo-climatic characteristics of the area of origin, rather than dietary values [26]. Our data shows an ^18^O-enrichment of raw cow milk ranging from 1.0 to 6.6‰ relative to that in the drinking water dependent on the season. Garbaras et al. [39] report a variation from 1–8‰ in the *δ*^18^O values between cow milk water and the drinking water. Ehtesham et al. [40] report an ^18^O-enrichment of approximately 4‰ in milk water compared to farm water, but no significant correlation between the two variables was found. Kornexl et al. [32] report an ^18^O-enrichment of 2–6‰ in milk water compared to ground water and other water sources.

There is usually no significant seasonal changes in *δ*^18^O values in groundwater (*δ*^18^O_GW_) due to its mean age typically covering decades to centuries. The distribution of the *δ*^18^O_GW_ values together with the mean recharge rates in the whole Slovenia is presented by Mezga et al. [41]. The *δ*^18^O_GW_ values of groundwater reported in our study ranged between −9.1‰ (Pannonian region) and −6.7‰, (Mediterranean region) with an average standard deviation within one year of 0.5‰.

The monthly distribution of *δ*^18^O_milk_ values, together with *δ*^18^O_GW_ values throughout the year 2012, is shown in Figure 5. The box plot of *δ*^18^O_milk_ values determined in June and December in 2013, 2014 and 2015 are also presented (Figure 5). Significant seasonal variations in the *δ*^18^O_milk_ values were observed, with higher values in summer days and lower values during winter. These findings support the fact that the water uptake by the cattle during the summer (at least in part) is from ingestion of fresh plants with water enriched in ^18^O as a consequence of evapotranspiration in leaves [23]. As previously discussed, body water is also strongly affected by temperature, which is related to the primary function of water in the thermoregulation of an animal’s body temperature [27]. A relationship between *δ*^18^O_milk_ and the season due to seasonal changes in the *δ*^18^O of forage plants, as well as in the body of the animal, linked to evapotranspiration was also reported by other studies [32,34,36,39,40].

The difference between the *δ*^18^O_milk_ and *δ*^18^O_GW_ values indicated that based on the isotopic composition of oxygen, it is possible to detect the addition of water to milk, i.e., with a greater certainty during the summer period (Figure 5). A simple experiment was performed to evaluate the capability to detect milk dilution with water. For this experiment samples of milk and GW from four locations covering three different regions were collected in May 2017: Mediterranean (Ajdovščina), Alpine (Črna na Koroškem, Selnica ob Dravi) and Pannonian (Ormož). The *δ*^18^O_milk_ values were the following −7.4‰, −6.0‰, −6.5‰ and −5.6‰, while *δ*^18^O_GW_ values were −9.3‰, −9.5‰, −10.0‰ and −10.2‰, respectively. A serial of dilution of a raw (authentic) cow milk with drinking water in the following volume percentages: 0%, 1%, 3%, 5%, 7%, 10%, 15%, 20% and 30% was performed.

The results presented in Figure 6 show that diluting milk with varying amounts of water decreases the *δ*^18^O_milk_ values. The correlation coefficient between *δ*^18^O_milk_ values and added water was high (R^2^ ≥ 0.89). Taking 2σ from determination of *δ*^18^O_milk_ as a maximum acceptable difference between *δ*^18^O_milk_ values in authentic and diluted milk the addition of >15% of water can be detected.

This experiment also shows that *δ*^18^O_milk_ values in raw milk and groundwater can provide a reference to detect adulteration and supports the findings of Lin et al. [42] who studied raw and manufactured milk from Taiwan. The method is more efficient than making cryoscopic measurements, especially when sodium chloride (NaCl) is added, which is a common practice, together with water to milk. The addition of NaCl can decrease the freezing point of water in the milk, which means that the dilution of water with cryoscopic method cannot be detected.

### 2.3. Lactose as an Internal Standard

Further, we check if it is possible to improve the detection of water addition using lactose and *δ*^18^O_lactose_ values as internal standard, since there is a close relationship between lactose synthesis and the amount of water drawn into milk [43]. Based on the findings from the European project [44], *δ*^18^O_lactose_ values are less affected by the season and relatively insensitive to changes in the cow’s diet. The difference for organically bound oxygen in lactose between regions is less pronounced than for oxygen of water, as oxygen-containing lactose is produced continuously over a longer time and therefore scrambling or exchange may occur. Also, *δ*^18^O_lactose_ values are enriched by approximately 25‰ relative to the cattle feeding water (Figure 7). This increase is related to the plant cellulose breakdown by the cattle and the incorporation of its glucose oxygen into lactose during synthesis. By adding water, the *δ*^18^O_milk_ changes accordingly, whereas the value of lactose does not change. It means that if *δ*^18^O_lactose_ and *δ*^18^O_milk_ are correlated, addition of water eliminates this correlation and can be detected. Thus, it seems that *δ*^18^O_lactose_ values could be used as internal standard to detect possible adulteration with water.

To test this hypothesis, we prepared a series of diluted milk samples and determined their *δ*^18^O_lactose_ values. Authentic milk samples were collected from the same locations as the first experiment covering the typical *δ*^18^O_GW_ values in Slovenia, to which 0%, 3%, 5%, 7%, 10%, 20% and 30% of water was added. The *δ*^18^O_lactose_ values for authentic samples ranged from 19.3‰ to 20.8‰. The lowest *δ*^18^O values of lactose were determined for Ajdovščina and the highest for Ormož. These data are comparable with the data obtained in other EU countries, for example in France (*δ*^18^O_lactose_ = 21.0 ± 1.8‰; *n* = 25), UK (*δ*^18^O_lactose_ = 21.3 ± 1.1‰; *n* = 36), Italy (*δ*^18^O_lactose_ = 16.8 ± 3.1‰; *n* = 55), and Spain (*δ*^18^O_lactose_ = 19.1 ± 2.1‰; *n* = 50), as reported in the final report of the European project [44].

The authenticity of the milk was evaluated by comparing the *δ*^18^O_lactose_ values with the corresponding *δ*^18^O_milk_ values of authentic and diluted milk samples. The results are presented in Figure 7 indicating a good correlation between *δ*^18^O_lactose_ values with the corresponding *δ*^18^O_milk_ of authentic samples (R^2^ = 0.98). For diluted samples, *δ*^18^O_milk_ is not more correlated with *δ*^18^O_lactose_ and falls in most of the cases outside the 95% confidence level of the regression line. Although the number of results is limited and this must be interpreted with care, it appears that it is possible to detect the adulteration with water even at lower amounts of added water (>7%). However, it should be pointed out that more research is needed on this topic especially since lactose, as the internal standard, may have its drawbacks in some cases. For example, higher measurement uncertainty is expected for milk with low lactose content.

## 3. Materials and Methods

### 3.1. Sampling and Sample Preparation

Authentic cow milk samples (*n* = 319) were collected directly from farms located in the four Slovenian macro-regions: Alpine, Dinaric, Pannonian, and the Mediterranean (Figure 8). The cow milk samples were obtained in summer (June) and winter (December) from 2013 to 2015. In 2012, samples of cow milk were collected monthly from January to December. In parallel, samples of groundwater (GW) were also collected. In addition, samples of goat (*n* = 15) and sheep milk (*n* = 22) were collected systematically during May, June and July in 2012 and 2013 from Bovec (Alpine), Karst (Dinaric), Vipava and Brkini (Mediterranean), and central Slovenian region (Dinaric). All samples were frozen and stored at −20 °C prior to analysis.

For pretesting the internal standardisation method, paired cow milk and drinking water samples from Alpine (Selnica ob Dravi, Črna na Koroškem), Mediterranean (Ajdovščina) and Pannonian (Ormož), were collected in May 2017. Samples were delivered with ice packs and then immediately registered and stored in the fridge upon receipt. All samples were stored at 4 °C for a period not exceeding 24 h.

### 3.2. Isolation of Casein

Analytical preparation of milk samples was carried out according to the standard procedure [21]. Fat was removed from milk sample of 25 mL by centrifugation (Type Centric 322 A, TEHTNICA, Železniki, Slovenia, 10 min at 3000× *g*) and the casein precipitated from the skimmed milk by acidification at pH 4.3 with 2 N HCl (CARLO ERBA, Val-de-Reuill, Loop, France) followed by subsequent centrifugation (10 min at 3000× *g*). The precipitate was rinsed once with pure water (Milli-Q system, Millipore Sigma, Burlington, MA, USA) and once with petroleum ether:ether (2:1) (both Merck, Darmstadt, Germany). After the centrifugation, sample was heated in a water bath (40 °C) until solvent was completely removed, and then freeze-dried. In parallel, the supernatant fractions and the washing water were combined and used for the next step—isolation of lactose.

### 3.3. Water Addition Experiment

Two different experiments with water addition were prepared. First, we prepared a series of authentic raw cow milk samples (V = 25 mL) from four locations covering three different regions: Mediterranean (Ajdovščina), Alpine (Črna na Koroškem, Selnica ob Dravi) and Pannonian (Ormož). diluted with different proportions of drinking water (0, 1, 3, 5, 7, 10, 15, 20 and 30% *v*/*v*). In this samples *δ*^18^O_GW_ and *δ*^18^O_milk_ values were determined. In the second experiment, we also prepared a series of authentic cow milk samples (V = 25 mL) from the same locations diluted with different proportions of drinking water (0, 3, 5, 7, 10, 20, and 30% *v*/*v*). In this experiment lactose was isolated and *δ*^18^O_milk_ values determined.

### 3.4. Isolation of Lactose

Lactose was obtained by heating the whey (supernatant) in a water bath (80 °C, 10 min) followed by filtration (Whatman 589/1, Sigma-Aldrich, St. Louis, MI, USA) and washing of the residue with not more than 5 mL of Milli-Q water [32]. The filtrate was then freeze-dried. Four replicates of each sample were prepared.

### 3.5. Determination of Stable Hydrogen and Oxygen Isotope Ratios

The determination of the stable of hydrogen and oxygen isotope ratios were performed using IRMS and expressed in the *δ*-notation in ‰ according to Equation (1) [45]:*δ*^i^E = (R(^i^E/^j^E)_sample_/R(^i^E/^j^E)_standard_) − 1(1)
where E is the element (H, O), R is the isotope ratio between the heavier “i” and the lighter “j” isotope (^2^H/^1^H, ^18^O/^16^O) in the sample and relevant internationally recognised reference standard. The delta values are multiplied by 1000 and expressed in units “per mil” (‰).The *δ*^2^H and *δ*^18^O were reported relative to the Vienna-Standard Mean Ocean Water (V-SMOW) standard [45].

The *δ*^18^O_milk_ and *δ*^18^O_GW_ values were determined directly in milk and water after equilibration with reference gas CO_2_/He (5% CO_2_) at 40 °C for 6 h. Measurements were performed on a continuous flow IRMS (GV Instruments ltd, Manchester, UK) connected with MultiFlow Bio preparation system (IsoPrime, GV Instruments Itd, Manchester, UK). The results for milk water were normalised against the following laboratory standards: W-3869 (seawater *δ*^18^O_VSMOW-SLAP_ = 0.36 ± 0.04‰) and W-3871 (snow water; *δ*^18^O_VSMOW-SLAP_ = −19.73 ± 0.02‰). An independent laboratory reference material W-3870 (Mili-Q water, *δ*^18^O_VSMOW-SLAP_ = −9.12 ± 0.04‰) was analysed periodically throughout the sequence as a control to ensure the quality of the results. The laboratory standards used are calibrated against certified reference materials: NIST 8535a- (Vienna Standard Mean Ocean Water 2; IAEA-VSMOW2) (water; *δ*^18^O_VSMOW-SLAP_ = 0.00 ± 0.02‰), RM 8537a- (Standard Light Antarctic Precipitation water; IAEA-SLAP2) (*δ*^18^O_VSMOW-SLAP_ = −55.50 ± 0.02‰) and NIST RM 8536 (Greenland Ice Sheet Precipitation water; GISP) (*δ*^18^O_VSMOW-SLAP_ = −24.76 ± 0.09‰). For each set of measurement, laboratory reference materials (W-3869 and W-3871) for normalization were measured four times; two times at the beginning of the batch, and two times at the end of the batch, while the control material (W-3870; MiliQ water) was measured six times; at the beginning, in the middle and at the end of the batch. Measurements precision was 0.1‰ for *δ*^18^O and 1‰ for *δ*^2^H.

The ^2^H/^1^H and ^18^O/^16^O and measurements of lactose and casein were performed at the Department of Food Quality and Nutrition, Research and Innovation Centre, Fondazione Edmund Mach in San Michele all’ Adige, Italy. The *δ*^2^H and *δ*^18^O values of lactose (*δ*^2^H_lactose_, *δ*^18^O_lactose_) and casein (*δ*^2^H_casein_, *δ*^18^O_casein_) were determined by transferring of freeze-dried samples, respectively, into a silver capsule and analysing the sample simultaneously using TC/EA pyrolyser (Thermo Finnigan, Waltham, MA, USA) coupled to a DELTA XP isotope ratio-mass spectrometer, IRMS (Thermo Scientific, Waltham, MA, USA). For normalisation of the results, two internal laboratory reference materials were applied: Caribou Hoof Standard (CBS) and Kudu Horn Standard (KHS). The sample weight was 0.2 mg and 0.4 mg for lactose and casein, respectively. The results for lactose and casein were calibrated against the following international reference materials: CBS keratin (Caribou Hoof Standard; *δ*^2^H_VSMOW-SLAP_ = −157.0 ± 0.9‰, *δ*^18^O_VSMOW-SLAP_ = +3.8 ± 0.3‰) and KHS keratin (Kudu Horn Standard; *δ*^2^H_VSMOW-SLAP_ = −35.3 ± 1.1‰, *δ*^18^O_VSMOW-SLAP_ = +20.3 ± 0.3‰). Measurements precision was ±0.2‰ for *δ*^18^O and ±1‰ for *δ*^2^H.

Data quality control charts were systematically recorded throughout the study period. To ensure the validity and comparability of the isotope results, the laboratory regularly participates in the Food analysis using Isotopic Techniques-Proficiency Testing Scheme FIT-PTS organized by EUROFINS (Nantes, France) three times per year. In this scheme, water and casein are also included.

### 3.6. Statistical Analysis

The data was processed using the statistical software package OriginPro 2018 (OriginLab, MicroCal Inc., Harrisburg, PE, USA), and Microsoft Excel (Microsoft Office Professional Plus 2019, Microsoft Corporation, Redmond, WA, USA). The existence of differences was verified through representation of variables within numeric data with box-plots (which graphically display summary of a data set: median, minimum, and maximum) or through regression analysis at a confidence level of 95%. One-way ANOVA was performed to determine the significant temporal (season, year) and spatial difference of variables. In the statistical test probability (*p*) values of less than 0.05 were used to indicate a significance level. If the significance was noted in a response factor, the calculation was followed by post-hoc testing using the Tukey’s Honestly Significant Difference (HSD) test.

## 4. Conclusions

This study demonstrated that by using the stable isotope composition of oxygen in milk water it is possible to discriminate milk from Slovenia according to the season and animal species, while regional discrimination is limited. The compositional differences in animal species indicate that diet and physiology have a strong control on animal isotope composition of body water and consequently also to milk. Seasonal variation in *δ*^18^O_milk_ values are controlled by evaporation processes. Actually, the “evaporation effect” may be related directly to the animal physiology as well as to ingestion of fresh grass with water enriched in ^18^O as a consequence of evapotranspiration in leaves. No significant statistical differences in *δ*^2^H_casein_ and *δ*^18^O_casein_ values according to the season and region of milk production was observed indicating that these two parameters provide more consistent isotopic signature with which to access the authenticity and origin of the milk. Further, the milk water is remarkably enriched in ^18^O compared to groundwater providing a possibility to detect addition of water. Based on our experiment it was found that >15% of added water can be detected by determining *δ*^18^O_milk_ and *δ*^18^O_GW_ values. The method using *δ*^18^O_lactose_ values as an internal standard was shown to be even more promising in improving the detection of the illegal watering of milk (>7%). A further improvement of this approach could be made in the future by analyzing higher number of samples originating from different countries.

## Figures and Tables

**Figure 1 molecules-25-04000-f001:**
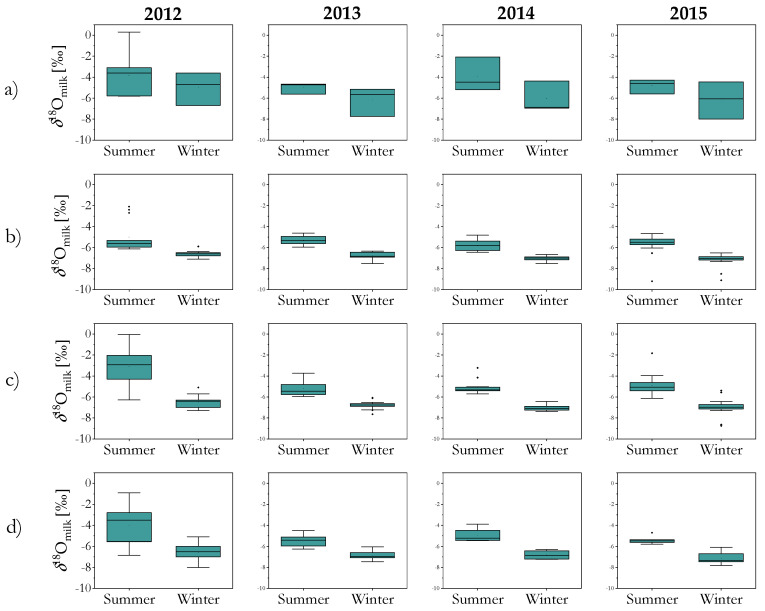
Box plots of the *δ*^18^O_milk_ values in cow milk collected from four macro-regions in Slovenia: (**a**) the Mediterranean, (**b**) Pannonian, (**c**) Alpine, (**d**) Dinaric region during summer and winter in 2012–2015.

**Figure 2 molecules-25-04000-f002:**
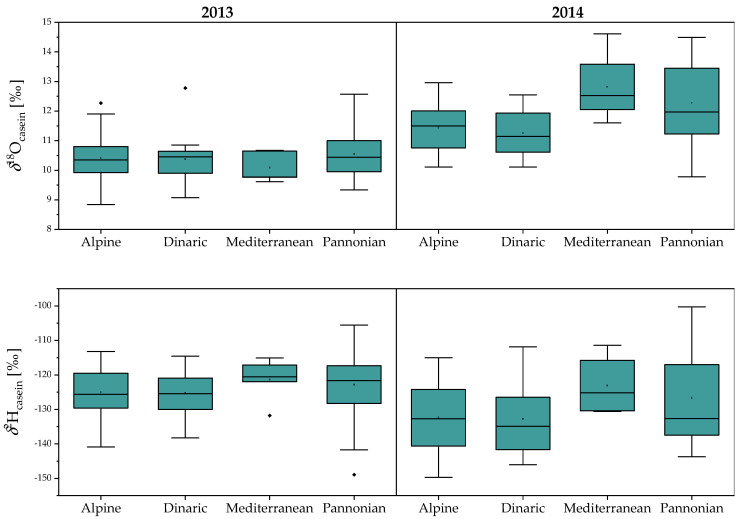
Box plots of *δ*^2^H_casein_ and *δ*^18^O_casein_ values in cow milk collected from four macro-regions in Slovenia in 2013 and 2014.

**Figure 3 molecules-25-04000-f003:**
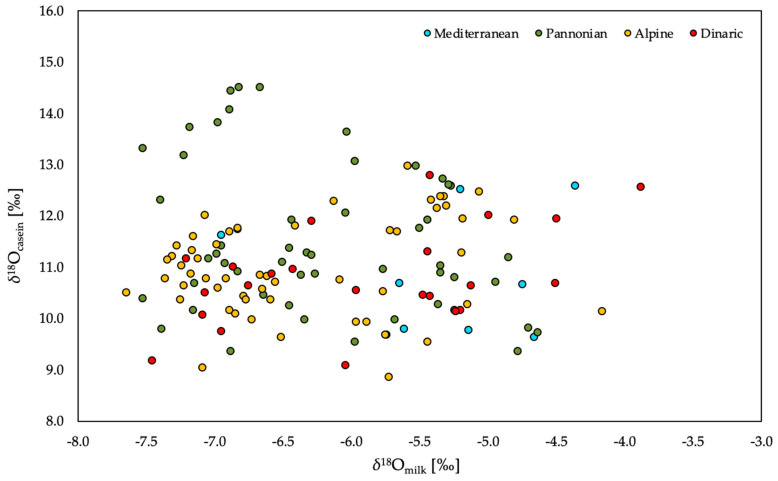
The relationship between *δ*^18^O_milk_ and δ^18^O_casein_ values in milk in relation to the region of production.

**Figure 4 molecules-25-04000-f004:**
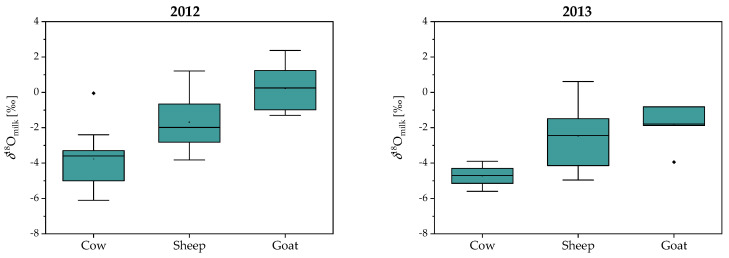
Box plot of the *δ*^18^O_milk_ values of different species (cow, sheep, and goat), collected from farms located in Mediterranean (Brkini, Vipava), Dinaric (Karst) and Alpine region (Bovec) in May and June in 2012 and 2013.

**Figure 5 molecules-25-04000-f005:**
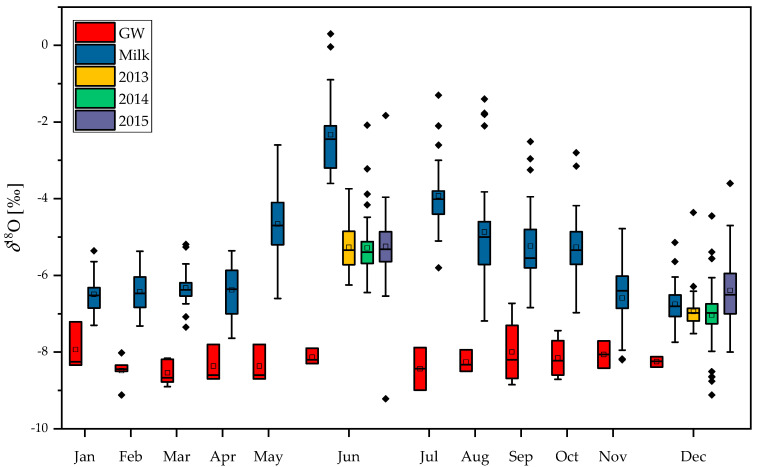
Seasonal variability in *δ*^18^O_milk_ and *δ*^18^O_GW_ values in 2012. For comparison, the box plot of *δ*^18^O_milk_ values determined in June and December relative to the year (2013–2015) are presented on the graph. Data presented are taken for all regions in Slovenia.

**Figure 6 molecules-25-04000-f006:**
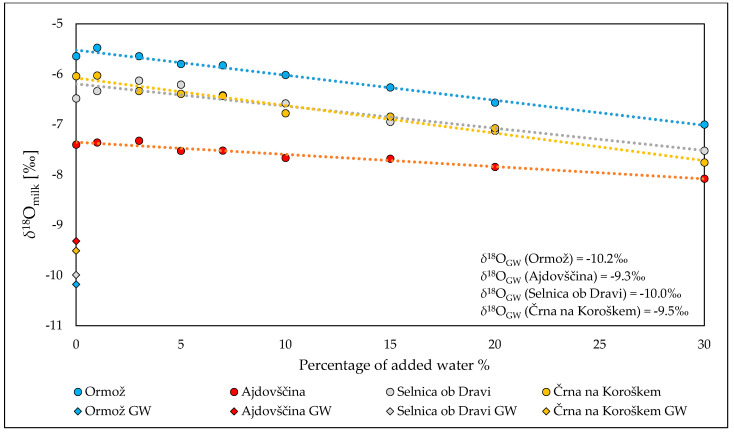
The relationship between *δ*^18^O_milk_ values and percentage of added water to authentic milk samples from different locations with different *δ*^18^O_GW_. Measured *δ*^18^O_milk_ are significantly (*p* < 0.05) related to the *δ*^18^O_GW_ in a regression analysis. The correlation coefficients between *δ*^18^O_milk_ values and added water were 0.98, 0.89, 0.98 and 0.96 for Ormož, Selnica ob Dravi, Črna na Koroškem and Ajdovščina, respectively.

**Figure 7 molecules-25-04000-f007:**
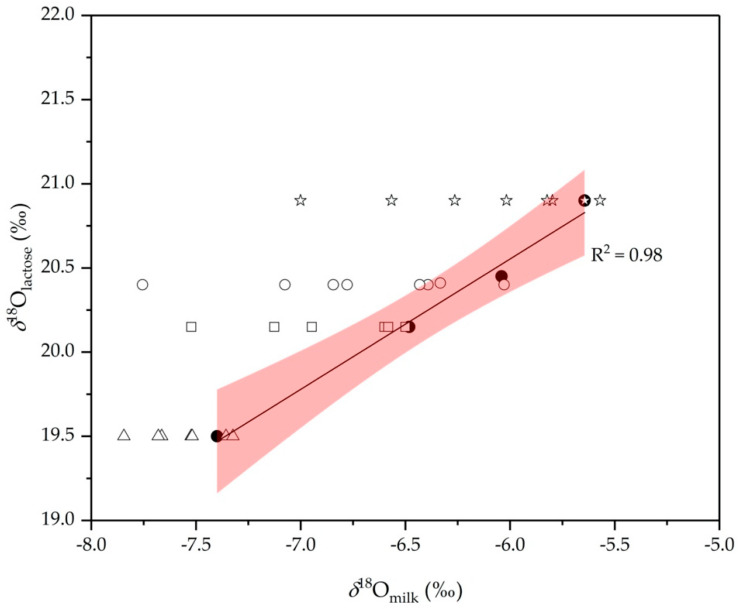
Relationship between *δ*^18^O_milk_ and *δ*^18^O_lactose_ values of authentic samples together with the regression line and 95% of confidence levels (R^2^ = 0.98). The data for diluted milk from Ajdovščina, Črna na Koroškem, Selnica pri Dravi in Ormož are also included. From right to left, points show the *δ*^18^O_milk_ when adding 3%, 5%, 7%, 10%, 20% and 30% of water to milk.

**Figure 8 molecules-25-04000-f008:**
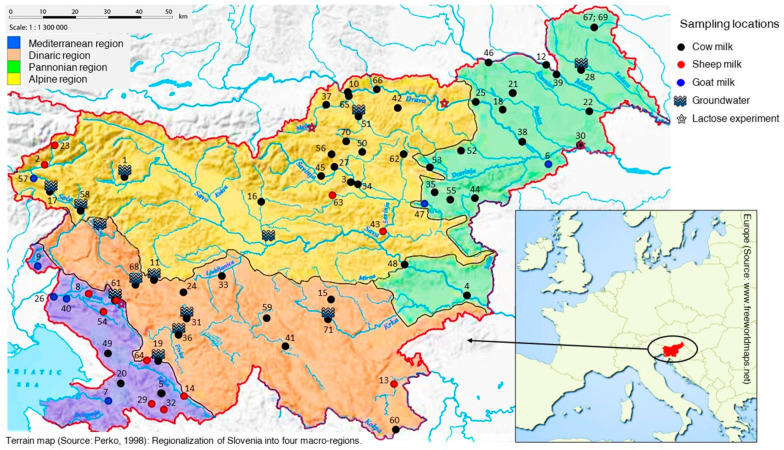
Macroregion division in Slovenia (after Perko, 1998) showing sampling locations of raw milk from different species covering geographical macro-regions (as indicated). The numbers correspond to the numbers of locations presented in Appendix A. The locations used in experiments are also presented.

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
