# Peer review of "Milk Authentication: Stable Isotope Composition of Hydrogen and Oxygen in Milks and Their Constituents"

_molecules, 2020, doi:10.3390/molecules25174000_

Round 1

Reviewer 1 Report

Revision of the manuscript entitled: Milk authentication: stable isotope composition of hydrogen and oxygen in milk and its ingredients by Gregorcic et al.

The manuscript deals with a stable isotope investigation on water of milk, casein and lactose. The aim was to identify the variations of δ18O depending on season, region and animal species (cow, goat and sheep); identify the correlation between δ18O in milk water and in casein of cow milk; define the influence of water addition on the isotope values.

General comments:

In general, the text is not well structured, the introduction contains a redundant part on the techniques used to measure the water content in milk. Furthermore, there is a clear misunderstanding in line 36: for Sudan it cannot be a problem of overpopulation. The differences between the milk water of the three animal species were never discussed. The isotopic values of the drinking water are not numerically indicated (it appears only in the drawing). It is not clear whether the three species of animals come from the same area and therefore potentially drink the same water. If the problem is not discussed, what does it do for milk authentication purposes? In the milk dilution experiments there are no values of the water used. If the water used for dilution is the water ingested by the cows, the initial enrichment, which is evident in three trends of figure 4, is not explained. The data for the added water are fundamental for the understand of the usefulness of the experiment. Their lack does not allow the achievement of the goal. The statements in the conclusions are not perfectly consistent with the results obtained: for instance, the discrimination of milk according to the seasons applies only to cows and only to Slovenia; it is not clear if the isotopic value of lactose never changes for milk from the same farm in the different seasons and it is not clear what the authors mean by "internal standard" (row 354); the conclusions do not report what has been achieved regarding the three objectives written in the introduction.

Particular comments

Line 149 add 2012 before 2013

Line 171 average values have little significance

Line 175 in this context what does “pedological” mean?

Figure 3 title of epsilon axis is not correct, it is referred to both milk and GW

Figure 4 regression line 95% confidence level: it is referred to the average of population or to the prediction of a new milk sample?

Line 307 check the equation

Line 316-317 The value 0,04 per mil and 0,02 per mil are standard deviation or standard error of the average values?

Author Response

See attachement.

Reviewer 2 Report

This is an interesting study that explores taxon-specific, seasonal, and regional differences in the stable isotope composition of milk and its constituents, with the aim of developing a methodology for identifying milk dilution with water.

General comments:

At times the passive voice is used here, and the writing is sometimes unclear. I have made suggestions throughout the manuscript (see attached) for clarifying wording, indicated by strike-throughs and replacement text (in red). In other cases, questions about specific parts of the text are highlighted and my question or comment is indicated in the same color.

What do the authors mean by d18Omilk, water? If this is whole milk, and not treated in any way, I suggest that the authors rename this to simply d18Omilk (as in Chesson et al. 2010).

References to articles cited in this review are at the bottom.

Title: I would reword this to “Milk authentication: Stable isotope composition of hydrogen and oxygen in the milk and its constituents”.

Line 27: I suggest removing the first sentence in the introduction. The second sentence is much more impactful and establishes the importance of this project!

Paragraph beginning at Line 44: I think a more effective start to the paragraph would be to outline

51-53: are there some citations that can be included here?

55-56: what do the authors mean by “minimum samples”?

Lines 51-75: I think this section is too long, and detracts from the point of the study. However, this is not my field of study, so I defer to other reviewers who are more familiar with this.

The manuscript should also have a background section on stable oxygen and hydrogen isotopes. Specifically, this section needs to cover how d18O and d2H values differ with respect to temperature, altitude, continentality, rainfall amount, aridity, etc. See reviews by Dansgaard (1964) and Gat (1996). This doesn’t have to be a long review—just a few sentences—but it should lay out some expectations in terms of how samples might differ with environment and seasonality. Next, the authors should include a few sentences on what factors contribute to differences in d18O among plants and animal species. This should include a mention of drinking behaviors, types of plants consumed, and physiological differences in fractionation (e.g. Barbour et al. 2007)

Results and Discussion Section: The authors need to include some information about the isotope data first. What were the standards that were used in the analyses? What is their expected d18O and d2H values? How many standards were run during the analysis, and what was the average value and standard deviation for each?

Lines 100-102: the authors should specify what statistical tests they conducted, and what the results were (e.g. df, p-value). I think they mean “Tukey test” instead of “Turkey”.

Lines 114-118: Here the authors should state what the d18O and d2H values are, rather than just referring to the figures and tables.

Line 127-129: I think it would be nice to include a figure comparing d18O of milk to that of casein across different regions, rather than just referring the reader to an SI table.

Figure 2 caption: should specify what regions these samples are from.

Line 144: citation?

Line 156-164: much of this should go in the background section (as I indicated above). In the discussion, the authors can delve more deeply into specifically what differences in diet and drinking behaviors exist among cattle, sheep, and goat. Here they can cite some other studies that show similar patterns. See for example, Kohn et al. 1996

Line 175: I don’t agree that d2H and d18O values would be influenced by pedological characteristics. It is true that soil characteristics can influence d15N values, but those are not studied in this paper.

Lines 177-178: need citation for groundwater recharge rates.

Lines 188-190: needs a citation for evapotranspiration and transpiration. For evapotranspiration, I suggest Barbour et al. 2007

Line 191: I’m not sure if reference 39 is an appropriate one here, as it does not discuss stable isotopes

Lines 193-195: This sentence should be a new paragraph (or move it to the beginning of the next paragraph), as the authors now discuss how d18O values can be used to detect dilution of milk.

Figure 3: Are these data from all regions? Is there any difference in groundwater d18O values by region?

Line 209: The authors should specify if these dilutions were carried out on summer or winter milk.

Line 212: The authors should note here the maximum and minimum difference between winter d18O values of milk and of groundwater, or at least refer to Figure 3.

Line 216: why is adding NaCl a common practice? What does it do to the milk?

Line 230: need citation for synthesis of lactose

Figure 4: I would like to see the range of d18O values of groundwater for each region plotted on this figure, so the reader can visually assess at what dilution the d18O values of milk begin to approximate those of groundwater. Also, the figure caption should specify that these are cow’s milk samples. Finally, the four analyzed regions are not clear. The authors should name them on the map.

Figure 5: The figure is missing a legend—what do the different marker shapes indicate? How is the regression calculated?

Line 240-243: The authors should specify that these are lactose d18O values; and they should include the citation number at the end of the sentence (“European Project” is to vague—what is the full name of this project?).

I would also like to see a table of the results of the dilution experiments, with d18O values of milk and of lactose. I was not provided either SI table, so I cannot check if the data are presented in either of those tables.

Figure 6: This map is a bit hard to read. I suggest that the authors include an inset that shows Slovenia in relation to the rest of Europe, for readers that are not very familiar with the geography of the region. This should also be the FIRST figure in the manuscript.

Equation 1: the formatting of “Ref” is a bit odd in the pdf.

Line 316: the authors should define the SLAP abbreviation when it first appears in the text. Also, I would like the authors to include how many measurements of each standard were conducted during the analyses.

Paragraph (lines 323-333): can the authors to include how many measurements of each standard were conducted during the analyses?

Line 350: specify that it is possible to discriminate milk in Slovenia.

References:

Abeni, F., Petrera, F., Capelletti, M., Dal Prà, A., Bontempo, L., Tonon, A., Camin, F., 2015. Hydrogen and oxygen stable isotope fractionation in body fluid compartments of dairy cattle according to season, farm, breed, and reproductive stage, PloS One 10, e0127391.

Barbour, M.M., 2007. Stable oxygen isotope composition of plant tissue: a review. Funct. Plant Biol. 34, 83–94.

Dansgaard, W., 1964. Stable isotopes in precipitation, Tellus 16, 436-468.

Gat, J.R., 1996. Oxygen and hydrogen isotopes in the hydrologic cycle, Annual Review of Earth and Planetary Sciences 24, 225-262.

Kohn, M.J., Schoeninger, M.J., Valley, J.W., 1996. Herbivore tooth oxygen isotope compositions: effects of diet and physiology. Geochim. Cosmochim. Acta 60, 3889–3896. https://doi.org/10.1016/0016-7037(96)00248-7.

Author Response

See attachement.
